# Biochemical analysis to study wild-type and polyglutamine-expanded ATXN3 species

**Grégoire Quinet**[1,2]*, **María Cristina Paz-Cabrera**[1], **Raimundo Freire**[1,2,3]*

**1** Unidad de Investigación, Hospital Universitario de Canarias, Instituto de Investigación Sanitaria de Canarias (IISC), La Laguna, Tenerife, Spain, **2** Instituto de Tecnologías Biomédicas, Universidad de La Laguna, La Laguna, Tenerife, Spain, **3** Universidad Fernando Pessoa Canarias, Santa María de Guía, Gran Canaria, Spain

* gregoirequinet@gmail.com (GQ); rfreire@ull.edu.es (RF)

**Data Availability Statement:** All relevant data are within the manuscript and its Supporting Information files.

**Funding:** This work was supported by grants PID2022-139691OB-I00 funded by MCIN/AEI/

## Abstract

Spinocerebellar ataxia type 3 (SCA3) is a cureless neurodegenerative disease recognized as the most prevalent form of dominantly inherited ataxia worldwide. The main hallmark of SCA3 is the expansion of a polyglutamine tract located in the C-terminal of Ataxin-3 (or ATXN3) protein, that triggers the mis-localization and toxic aggregation of ATXN3 in neuronal cells. The propensity of wild type and polyglutamine-expanded ATXN3 proteins to aggregate has been extensively studied over the last decades. *In vitro* studies with mass spectrometry techniques revealed a time-dependent aggregation of polyglutamine-expanded ATXN3 that occurs in several steps, leading to fibrils formation, a high status of aggregation. For *in vivo* experiments though, the techniques commonly used to demonstrate aggregation of polyglutamine proteins, such as filter trap assays, SDS-PAGE and SDS-AGE, are unable to unequivocally show all the stages of aggregation of wild type and polyglutamine-expanded ATXN3 proteins. Here we describe a systematic and detailed analysis of different known techniques to detect the various forms of both wild type and pathologic ATXN3 aggregates, and we discuss the power and limitation of each strategy.

## Introduction

Polyglutamine (polyQ) disorders are associated with a pathological form of a protein harbouring an interrupted tract of polyQ, resulting from an expanded (CAG)n trinucleotide repeat within the coding region of the associated gene [1]. One of these proteins is Ataxin-3 or ATXN3 (UniProt ID P54252), an evolutionary conserved protein of a theoretical length of 42 kDa from the family of cysteine proteases, that harbours, among others, a Josephin Domain and a polyQ repeat [2, 3]. The length of the polyQ in the normal form of ATXN3 (wild type, ATXN3 WT) ranges from 12 to 44 repeats, while the pathological variant (ATXN3 polyQ) expands from 55 to 87 repetitions [4]. ATXN3 polyQ is the pathogenic protein associated with spinocerebellar ataxia type 3 (SCA3), a dominantly inherited neurodegenerative disorder also called Machado-Joseph disease [5]. As in other polyQ disorders, expansion of the glutamine tract in ATXN3 directly correlates with the severity of the disease [3]. Indeed, ATXN3 polyQ

10.13039/501100011033 and ERDF A way of making Europe (European Union) to RF and grant OA23/071 from Fundación DISA to GQ.

**Competing interests:** The authors have declared that no competing interests exist.

can oligomerize and aggregate leading to toxic intraneuronal inclusions that constitute the pathological hallmark of SCA3 [5–7].

Although not fully understood, the aggregation features of ATXN3 proteins have been extensively investigated over the past decades. A mass spectrometry study of the aggregation products obtained *in vitro* with purified recombinant ATXN3 proteins, proposed a four-step mechanism for the aggregation of ATXN3. This mechanism includes an early oligomerization phase, slow for ATXN3 WT but accelerated for ATXN3 polyQ, followed by a late oligomerization phase, that then leads to the formation of a higher order of aggregation, called protofibrils. ATXN3 polyQ proteins finally aggregate irreversibly into SDS-resistant fibrils in a process dependent on large polyQ tract [8]. Several other *in vitro* studies support this multi-stage mechanism during which both WT and polyQ versions of ATXN3 aggregate, with faster aggregation and additional formation of SDS insoluble aggregates for the pathological variants [9–11].

Conventional biochemical methods used to study ATXN3 forms *in vivo* include SDS-Polyacrylamide Electrophoresis (SDS-PAGE) [12–15] and filter trap assay (FTA) [12, 15–17]. However, these techniques, when used individually, are not able to distinguish the different steps of aggregation of ATXN3 in detail. Furthermore, preparation of the sample containing ATXN3 aggregates before the analysis is a critical step for analysis as conventional lysis methods eliminate the pelleted cellular debris in which insoluble aggregate proteins might remain. Lysis buffers might additionally lack sufficient detergent for effective solubilization therefore affect the detection of late stages ATXN3 aggregates.

Since the degree of aggregation is intimately linked to the SCA3 disease pathogenesis, the capability to study all forms of ATXN3 aggregates through simple and reproducible techniques *in vivo* is of great interest. Nevertheless, the multi-step aggregation of ATXN3 is time-dependent and can be influenced by several parameters, including the biological model used, the type of ATXN3 isoforms and the amount of expressed proteins [8, 11, 12]. Here we describe a cost-effective and straightforward biochemical method to separate and resolve monomers, oligomers and high molecular weight (HMW) species from WT and pathological ATXN3 variants. By optimizing the time of protein expression, using a modified protein fractionation procedure previously described for Huntingtin polyQ proteins [18], and refining parameters of SDS-PAGE, SDS-Agarose Gel Electrophoresis (SDS-AGE) and FTA assays, this method enables a clear visualization of ATXN3 intermediates in cellular models.

## Methods

The protocol described in this article is published on protocols.io *dx.doi.org/10.17504/protocols.io.dm6gpz9q8lzp/v1* and is included for printing as S1 File with this article.

## Expected results

ATXN3 WT and polyQ expanded variants, fused with a protein tag or a fluorescent protein to ease detection, have been widely used as a molecular tool to express the protein of interest in cellular models for investigating the SCA3 pathogenesis [16, 19, 20]. A straightforward cellular model often chosen to express ATXN3 proteins for their study *in vivo* is the embryonic kidney HEK-293T cell line [12, 15, 16]. In this study, we developed and optimized protocols applied to HEK-293T cells transiently expressing a GFP-tagged full length human ATXN3 WT with 23 repeats of glutamine [20], as well as ATXN3 with a polyQ tract containing an 80-glutamine tract [19]. To optimize the expression of the protein, we first transfected different amount of the GFP-ATXN3 WT expressing plasmid (S1A Fig). In our hands, expression of ATXN3 was

optimal using 3 μg of plasmid, which was subsequently used in all following experiments (S1 File).

Given the diverse forms and solubilities of ATXN3 species, a biochemical fractionation was performed to isolate the different types of ATXN3 aggregates [21]. For this, a protein fractionation originally developed for the polyQ protein Huntingtin [18] was adapted (Fig 1). First, a "soluble buffer" containing the non-ionic detergent Triton X-100 was used to lyse cells and separate soluble forms of ATXN3 by high-speed centrifugation. Because ATXN3 can bind to chromatin [19], a nuclease was added to the lysis buffer to prevent sedimentation of soluble ATXN3 species with chromatin and remove nucleic acids possibly entangled around aggregates [22]. Non-soluble proteins were then resuspended in a buffer containing 4% SDS ("insoluble buffer"), followed by sonication and boiling of the sample.

Separation of proteins into two fractions enables a better resolution of ATXN3 species, providing crucial information about their solubility and thus insights into the progression of ATXN3 aggregation. Monomer and early polymers of ATXN3 are separated in the soluble fraction after centrifugation, while Triton-insoluble ATXN3 forms solubilized in SDS are retained in the second fraction. ATXN3 polyQ SDS-resistant fibrils from the final stage of aggregation were expected to be found in the insoluble fraction since they are highly stable in ionic detergent [9, 23]. We thus tried several buffers containing either SDS, DTT, urea or a combination of these compounds, to resuspend ATXN3 variants from the insoluble pellet. Interestingly, the 7M urea buffer and the mix buffer (4% SDS, 100mM DTT, 7M urea) efficiently solubilized the ATXN3 polyQ HMW forms that remained blocked in the stacking gel when the insoluble fraction was resuspended in 4% SDS only (S1B Fig). Therefore, to preserve ATXN3 polyQ HMW forms for visualization, which are likely ATXN3 polyQ fibrils, the buffer containing 4% SDS is the most suitable (S1B Fig) and is used in the fractionation protocol of soluble and insoluble ATXN3 forms detailed in S1 File.

To subsequently analyse the different ATXN3 species separated by fractionation, SDS-PAGE was used first. The SDS-PAGE settings were optimised for the maximal resolution of all ATXN3 species. Fifteen micrograms of soluble and insoluble fractions were run on an 8% polyacrylamide separating gel with a 4% polyacrylamide stacking gel. ATXN3 SDS-resistant aggregates behave as HMW proteins in SDS-PAGE and remain in the stacking of polyacrylamide gel, as previously described [23–25]. The entire gel, including the stacking and the wells was therefore transferred onto a membrane under high transference conditions to allow HMW proteins to transfer. Due to their size and solubility, the ability of these large insoluble aggregates to be charged during the electrophoresis process is limited and could face difficulty to efficiently enter the 4% polyacrylamide stacking gel (S1C Fig). We observed that increasing the electrophoresis voltage improved the migration of the ATXN3 species in the stacking gel (S1C Fig). As a control for the fractionation and analysis by SDS-PAGE, His-tagged ATXN3 WT was expressed, purified and subsequently processed with this protocol. As expected, the ATXN3 WT was only present in the soluble fraction and run as a monomer, demonstrating that the protocol does not result in artifacts, at least with the soluble ATXN3 WT, thereby strongly suggesting that the non-monomeric forms observed after fractionation of cell extracts are the result from aggregation (S1D Fig).

We hypothesized that the signal in the blot from the stacking gel could correlate with the amount of aggregates present in the loaded samples. When loading increasing amounts of the SDS-resistant extracts, a linear correlation with ATXN3 polyQ HMW signal was observed. This indicates that under these SDS-PAGE conditions, the signal of the HMW proteins is quantitative up to the loading of 12 μg of total protein (S1E Fig), although this amount could vary depending on the quantity and the quality of the aggregates formed in the cellular conditions used.

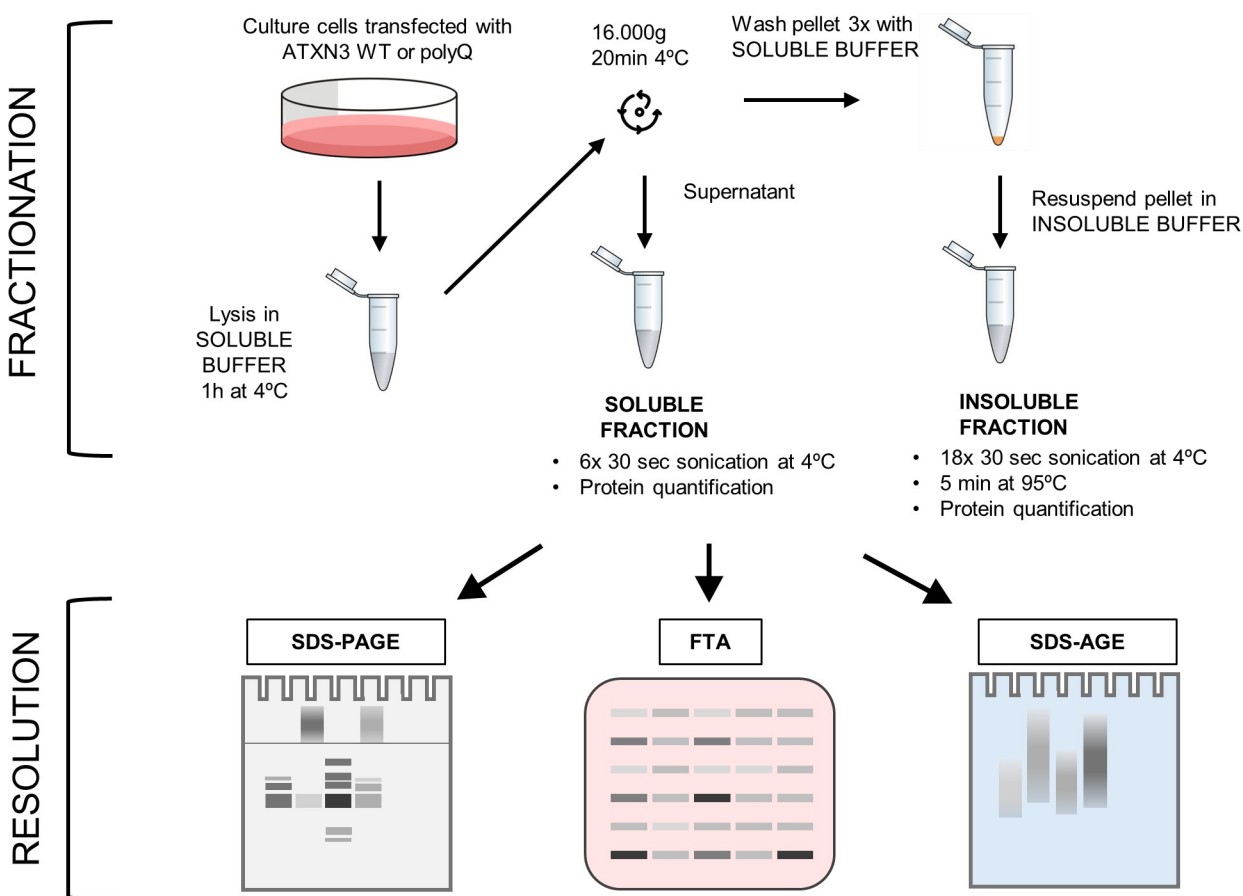

**Fig 1. Experimental workflow of the fractionation assay.** Cultured cells expressing ATXN3 proteins are washed in PBS and lysed in soluble buffer. After centrifugation, the supernatant is saved as the soluble fraction. Pellets are washed with soluble buffer and resuspended with insoluble buffer, hereafter referred as the insoluble fraction. Fractions are sonicated, incubated for 5 min at 95˚C for insoluble fractions, before quantification. Samples are then analysed by Polyacrylamide Gel Electrophoresis (SDS-PAGE), Filter Trap Assay (FTA) and SDS-Agarose Gel Electrophoresis (SDS-AGE) with the optimized parameters described in this article, to allow an optimal resolution and quantification depending on the ATXN3 species of interest.

Given that the multi-step aggregation dynamics of ATXN3 polyQ is a time-dependent process influenced by various parameters [8, 11, 12, 26], our analysis was carried out at different times post transfection to optimize the formation of ATXN3 aggregates in our cellular model. Cells transiently expressing GFP-ATXN3 polyQ were harvested at different times from 1 to 7 days post-transfection, subjected to the biochemical fractionation previously mentioned and run on SDS-PAGE (S2A Fig). The GFP signal from the HMW SDS-resistant fibrils was observed in the stacking gel starting from samples collected at 3 days post-transfection, but with a greater amount observed 6 and 7 days after transfection (S2A Fig and Fig 2A). As in our experimental setup no SDS-resistant GFP-ATXN3 polyQ fibrils were observed at 2 days post-transfection, whereas maximal of aggregation was detected at day 7 day post-transfection, subsequent experiments were carried at day 2 and 7 post-transfection as representative time points in the ATXN3 aggregation process. As expected and consistent with previous findings, HMW SDS-resistant aggregates were not detected in extracts of cells transfected with GFP-ATXN3 WT, even after 7 days of transfection (Fig 2A) [11, 25].

Interestingly, we observed more ATXN3 aggregation intermediates/oligomers in the soluble fractions of both WT and polyQ GFP-ATXN3 at longer times after transfection (Fig 2A).

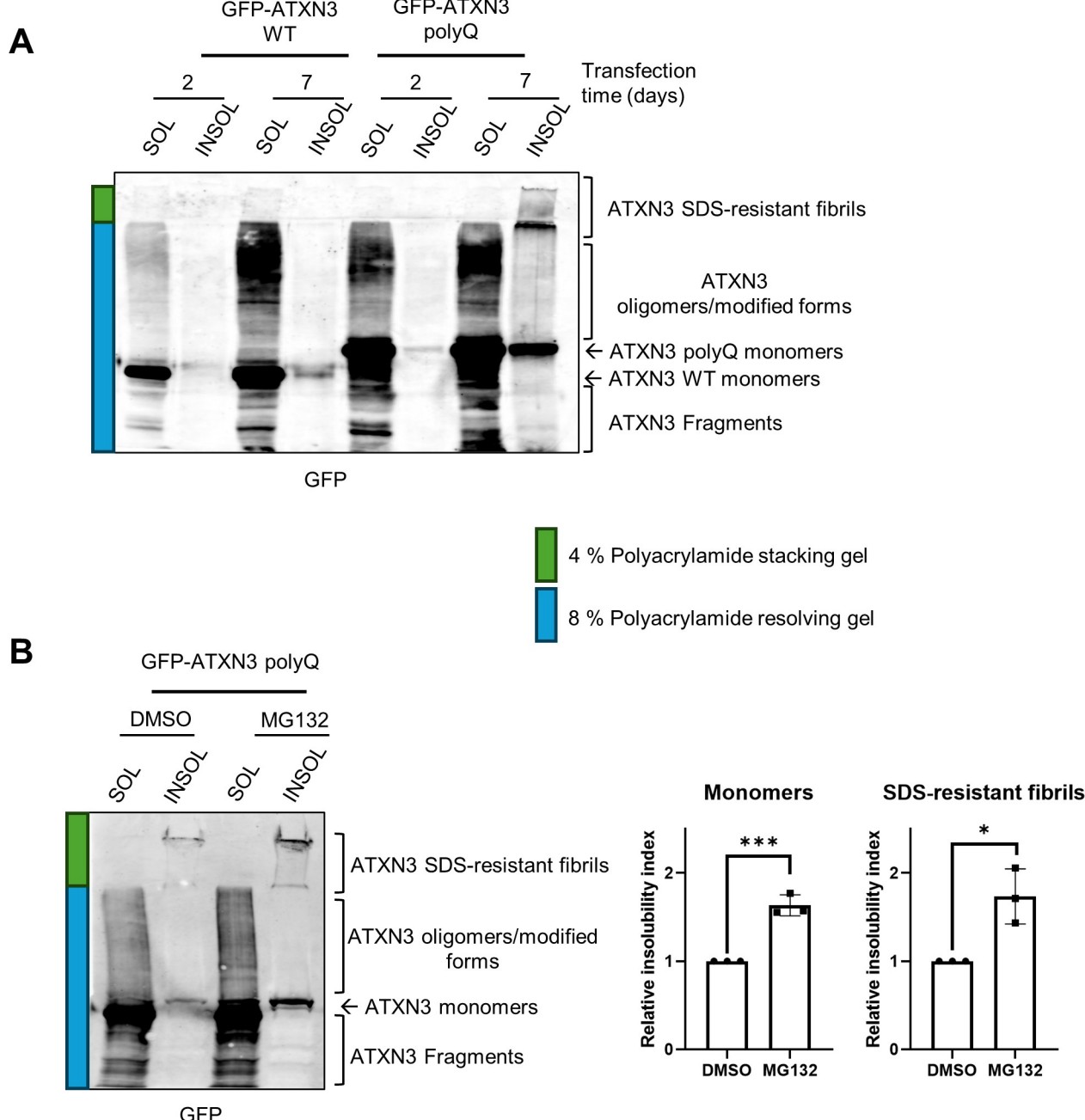

**Fig 2. SDS-PAGE allows the resolution of ATXN3 monomers, polymers/modified forms and HMW fibrils.** (A) HEK-239T were transfected with GFP-ATXN3 WT or GFP-ATXN3 polyQ plasmids at the indicated times cell were collected and subjected to the fractionation and SDS-PAGE as described in the protocols section. The complete gel including the stacking gel was transferred onto nitrocellulose followed by western blotting using a GFP antibody. In green and blue are indicated parts corresponding to the separating and stacking phases. (B) HEK-239T were transfected with ATXN3 polyQ and 7 days later treated with DMSO or MG132 (5 μM) overnight. Cells were then lysed and processed as in (A) (left panel). Quantification of the insoluble GFP-ATXN3 polyQ, in monomers (middle panel) and HMW aggregates (right panel) (n = 3). The relative ratio between the insoluble and soluble fraction was calculated and called insolubility index. The relative insolubility index of DMSO-treated samples is set to 1. Student's t-test was used to determine the statistical significance. Error bars represent the standard deviation. * p<0.5, ** p<0.01 and *** p<0.001.

More ATXN3 WT and polyQ monomers were additionally detected in the insoluble fraction after 7 days of transfection (Fig 2A). This suggests that earlier steps of both WT and polyQ ATXN3 aggregation keep occurring after several days of transfection, possibly due to the new synthesis of ATXN3 proteins. We also noticed that the levels of GFP-ATXN3 polyQ were somewhat higher than those of GFP-ATXN3 WT, especially noticeable at the soluble form. This could be due to differences in the transfection and/or expression process, even though the same amount of plasmid was transfected. Alternatively, this could reflect a higher protein stability of GFP-ATNX3 polyQ compared to ATXN3 WT.

According to the so-called toxic fragment hypothesis, the proteolytic cleavage of ATXN3 by cellular proteases such as calpains plays a crucial role in its toxicity during SCA3 pathogenesis [27]. This hypothesis proposes that the generation of cleaved ATXN3 polyQ proteins initiates the aggregation process associated with cellular dysfunction in the disease [23, 24]. Interestingly, using an anti-GFP antibody, we were able to detect proteins with smaller molecular weight than monomeric GFP-ATXN3 versions of the proteins in the soluble samples after fractionation (indicated as ATXN3 Fragments in Fig 2A and 2B). Size and abundance of these fragments varied between samples transfected for 2 and 7 days, suggesting that the duration of protein expression might have an impact on the proteolysis of ATXN3 proteins (Fig 2A and 2B). As some information could be missed in this analysis using the anti-GFP antibody, the fractionated ATXN3 polyQ samples were studied by western blot using different antibodies. We used an antibody against full length ATXN3, another against the C-terminal part of ATXN3 and finally, an antibody that recognizes polyglutamine tracks (S2B Fig). The patterns resulting from the westerns of the 3 antibodies were comparable to the patterns obtained using anti-GFP. The only exception was that the two ATXN3 antibodies and the polyglutamine antibody recognized some proteins smaller than the GFP ATXN3 polyQ monomer, that were not recognized by the anti-GFP antibody (Fig 2A and S2B Fig). These bands could either correspond to overexpressed GFP-ATXN3 polyQ that was processed to smaller fragments without GFP or to (modified forms of) endogenous ATXN3. Indeed, endogenous ATXN3 migrated with the same mobility as bands observed in the soluble and insoluble fractions of cells expressing GFP-ATXN polyQ when analysed with the different ATXN3 antibodies (S2C Fig). Therefore, the use of different antibodies against ATXN3 is advisable for the detection of protein fragments, and care should be taken when drawing conclusions regarding a model with both endogenous and exogenous proteins. Since the patterns of the blots with the 4 antibodies employed in this study were similar, we decided to use the anti-GFP antibody as it only detects the exogenously expressed polyQ expanded protein of our interest.

Notably, other research groups reported HMW SDS-resistant aggregates after only 2 or 3 days of transfection in cells transiently expressing tagged ATXN3 polyQ [15, 16]. These discrepancies observed in the kinetics of the last stage of ATXN3 polyQ aggregation could be caused by various factors. First, the aggregation of ATXN3 may vary depending on the cellular model used, as a non-exhaustive list of cellular factors such as chaperones, interacting partners, or post-translational modifications has been shown to modulate ATXN3 aggregation [28, 29]. The dynamic of ATXN3 aggregation additionally depends on protein concentration thus the construct used and the efficiency of protein expression will influence the timing of aggregation formation [26]. Also, the length of the polyQ tract carried by the protein influences ATXN3 aggregation dynamics [9, 28].

Several studies have demonstrated that the ubiquitin proteasome system is affected in SCA3 and it is known that proteasome activity regulates the ATXN3 aggregation process [30–32]. We therefore investigated the impact of proteasome inhibition on ATXN3 aggregation in our setup. Cells expressing GFP-ATXN3 polyQ for 7 days were treated with MG132 overnight, and samples were subjected to the fractionation and analysis by SDS-PAGE (Fig 2B). The

relative insolubility index, determined by quantification of the insoluble vs soluble fractions, demonstrated that proteasome inhibition led to accumulation of ATXN3 insoluble monomers and HMW forms, which is in agreement with previous reports [30–32].

Together these results show that cell fractionation followed by SDS-PAGE with adapted settings is an efficient method to study and quantify different ATXN3 variants, including ATXN3 fragments, monomers, polymers, modified forms and HMW SDS-resistant fibrils. However, SDS-PAGE analysis is not suitable to resolve protein species larger than 500 kDa, primarily due to the limitation of polyacrylamide gel pore size, but also, as in our case, due to the poor solubility of HMW SDS-insoluble ATXN3 aggregates that remain in the stacking gel after electrophoresis. We therefore employed the Filter Trap Assay (FTA), a technique proposed for visualizing large protein aggregates. In this assay, vacuum pressure is applied to a cassette forcing samples through a filter membrane where only particles larger than a certain pore size are retained and can be visualized [10, 12, 15–17]. We adapted an existing FTA protocol for global aggregated proteins in human cell lines using a 0.2 µm pore size acetate membrane to analyse ATXN3 aggregates (see S1 File for details) [33].

Cell expressing GFP-ATXN3 WT or polyQ for 2 and 7 days were fractionated and analysed by FTA (Fig 3A). Consistent with our observations after SDS-PAGE in Fig 2A and S1D Fig, GFP signal from HMW SDS-resistant forms was only detected in the insoluble fraction of day 7 GFP-ATXN3 polyQ samples (Fig 3A and 3B). Furthermore, positive signal for ATXN3 aggregates was observed when insoluble precipitates were resuspended in 4% SDS, and in a lesser extent with DTT or urea (S3A Fig). The absence of signal in the sample resuspended in the mix buffer (4% SDS; 100 mM DTT; 7M urea) suggests that HMW ATXN3 species were completely solubilized in this condition, as observed in S1B Fig. These results indicate that the trap assay effectively retains the HMW ATXN3 polyQ species. Additionally, when cells were treated with the proteasome inhibitor MG132, more aggregates in the insoluble fraction retained in the FTA filter (Fig 3B), suggesting that proteasome inhibition caused accumulation of ATXN3 polyQ aggregation into HMW fibrils, as observed in Fig 2B and expected from previous studies [30–32]. To ensure that FTA assays are quantitative, the filter membrane must remain unsaturated to allow samples to pass through without obstructing the liquid flow [12, 15]. Indeed, a strong linear correlation between the GFP signal detected and the quantity of total protein loaded with FTA was measured (S3B Fig). This result supports the suitability of this assay to visualize and quantify ATXN3 polyQ aggregates of over 0.2 µm (the size of the membrane pores) when up to 40 µg of total protein is loaded, although this amount could vary depending on the aggregate quantity in the samples.

Although SDS-PAGE and FTA allow a global quantification of SDS-insoluble HMW of ATXN3 polyQ, neither of these assays are suitable for the separation of the HMW ATXN3 species by size. We therefore employed SDS-Agarose Gel Electrophoresis (SDS-AGE), which had been proposed as straightforward technique to separate fibrous protein aggregates, and is routinely used to screen amyloid aggregates [34]. SDS-AGE was used additionally to investigate the large fibrils oligomers of the polyQ protein Huntingtin [35, 36] and more recently also to separate HMW ATXN3 [37, 38]. Similar to SDS-PAGE, this method relies on the ability of the protein substrates to enter the polymerized gel and to distribute according to their molecular weight and their resistance to detergent. However, unlike polyacrylamide gels, the pore size of an agarose gel of 1% has been determined to be around 100 nm [39], theoretically capable to resolve proteins exceeding 1 million kDa [40].

To resolve the HMW ATXN3 species observed in the insoluble fractions after fractionation described in this paper, we adapted a protocol of agarose gel electrophoresis (Fig 4) (see protocol section S1 File) [40, 41]. As in SDS-PAGE and FTA, samples from cells expressing GFP-ATXN3 WT and polyQ were collected and fractionated, after which the insoluble

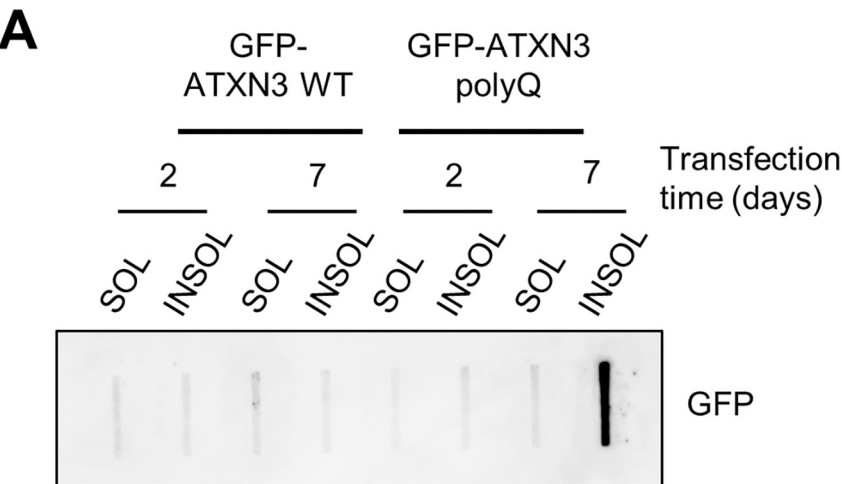

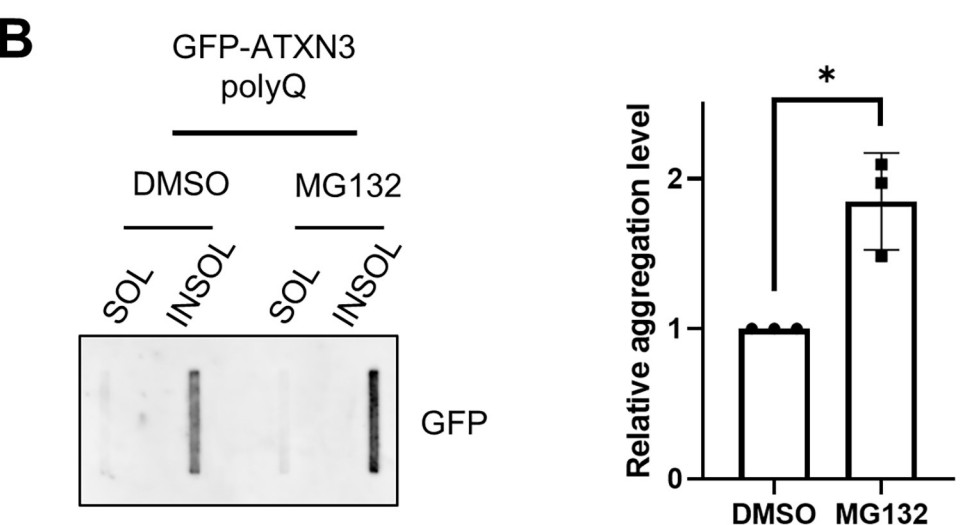

**Fig 3. FTA analysis of ATXN3 aggregates.** (A) HEK-239T transfected with the indicated expression plasmids were collected at the indicated times and subjected to the fractionation and FTA protocols described in S1 File. The membrane was subsequently immunoblotted with anti-GFP antibody. (B) HEK-239T cells expressing GFP-ATXN3 polyQ for 7 days were treated with DMSO or 5 μM MG132 overnight, after which cells were collected and analysed as in A (left panel). The relative GFP signal from the insoluble fractions was quantified (n = 3) and normalized to the signal of the DMSO-treated cells (right panel). Student's t-test was used to calculate the statistical significance. Error bars represent the standard deviation. * p<0.05.

fractions containing the large SDS-resistant ATXN3 species were analysed by SDS-AGE. HMW aggregates were resolved in the agarose gel, as GFP blotting reveals a smear of which the intensity and size can be compared between conditions (Fig 4A and 4B). More HMW ATXN3 aggregates were observed with ATXN3 polyQ at 7 days after transfection (Fig 4A), which is consistent with the previous results observed with SDS-PAGE and FTA (Figs 2A, 3A and S2A Fig). The resolution of HMW species above 250 kDa by SDS-AGE revealed that larger ATXN3 aggregates form 7 days after transfection, mostly in ATXN3 polyQ expressing cells and to a lesser extent in cells expressing the WT version of ATXN3. Furthermore, after MG132 treatment a stronger GFP signal of higher molecular weight species was observed,

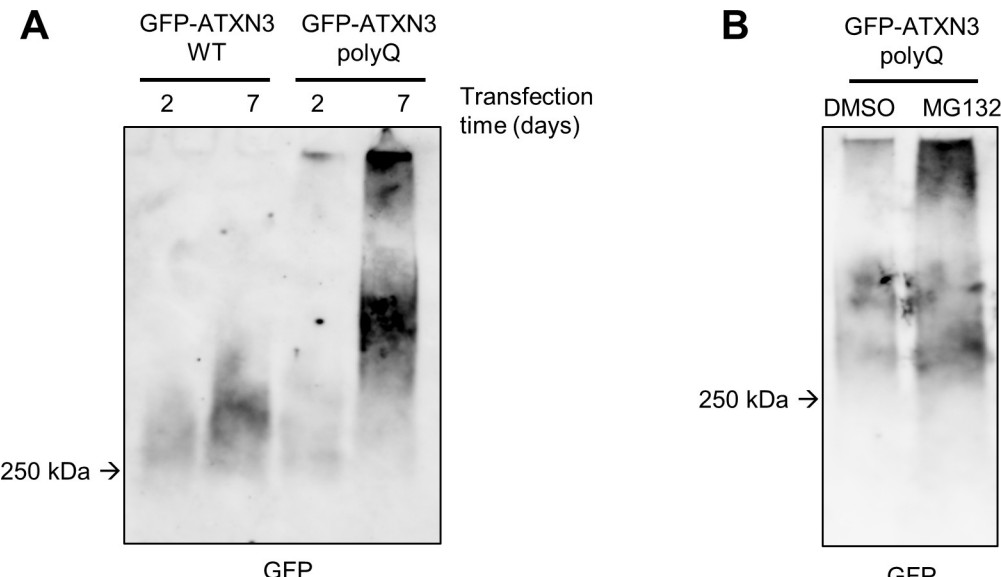

**Fig 4. Adapted SDS-AGE for resolving HMW ATXN3 aggregates.** (A) HEK-239T were transfected with the indicated expression plasmids and collected at the specified times before fractionation, SDS-AGE and immunoblotted with an anti-GFP antibody. (B) As in (A) with cells transfected with GFP-ATXN3 polyQ for 7 days and treated with DMSO or 5 μM MG-132 for 16 hours.

implying that proteasome inhibition promotes accumulation of such larger aggregates (Fig 4B).

Altogether these findings highlight that SDS-AGE is an appropriate method to visualize the different sizes of HMW ATXN3 aggregates. The limitations of SDS-AGE are evident in its challenging quantitative aspect, as the migration of proteins is not well-defined and is influenced by various parameters previously described [42]. However, compared to SDS-PAGE and FTA, SDS-AGE offers additional insights into the distribution of ATXN3 aggregates based on their molecular weight, providing crucial information for researchers investigating the formation of large ATXN3 polyQ fibrils.

It is worth noting that we observe an increase in ATXN3 polyQ and WT aggregation in protein samples from soluble fractions that were heated at more than 40°C or were left at room temperature for more than 24 hours, as a white precipitate was formed that led to a positive signal for HMW proteins when analysed by SDS-PAGE, FTA or SDS-AGE (S3C Fig). This suggests that soluble WT and polyQ ATXN3 monomers/oligomers can artificially aggregate under these conditions, similar to reported for the Huntingtin polyQ protein [18]. It is therefore recommended to avoid several thawing cycles, for example by freezing samples in aliquots, in addition to thawing the fractions on ice to prevent *in vitro* aggregation after extraction.

In conclusion, our results show that, when possible, optimization of protein expression time is required to study WT and polyQ-expanded ATXN3 aggregation *in vivo*. Additionally, an important step in the analysis is the biochemical fractionation that enables an optimized solubilization of ATXN3 species. Among the techniques to visualize ATXN3 intermediates described here, SDS-PAGE can be used as a quantitative method for all kinds of ATXN3 forms, although it is not able to separate HMW SDS-resistant fibrils. FTA is a straightforward and reproducible method, but only allows to quantify ATXN3 aggregates larger than 0.2 μm, while SDS-AGE is well adapted to separate and qualitatively analyse HMW ATXN3 species

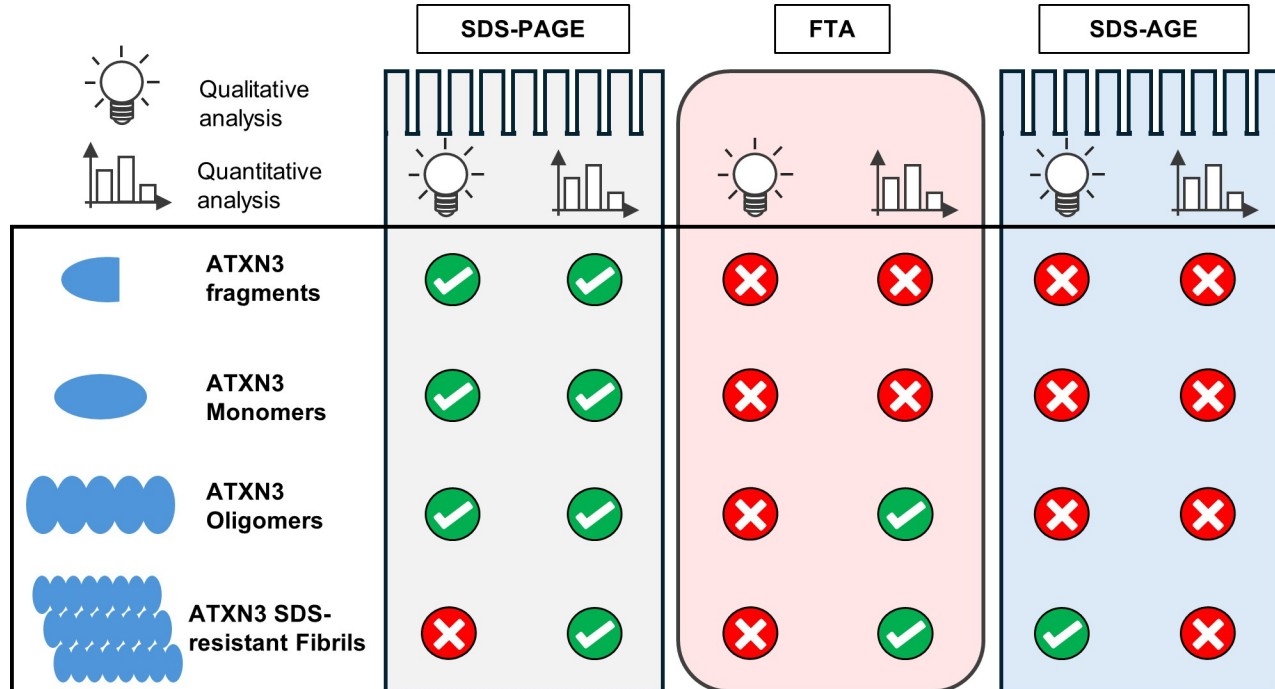

**Fig 5. Analysis of ATXN3 forms by SDS-PAGE, FTA and SDS-AGE.** Table illustrating the different options for qualitative and the quantitative analysis of ATXN3 forms by SDS-PAGE, FTA and SDS-AGE assays. Note that FTA is only suitable for aggregates superior to 0.2 μm.

(Fig 5). Therefore, depending on research context and on the ATXN3 species targeted for analysis, one or a combination of these methods can be used to accurately characterize ATXN3 aggregation (Fig 5). This protocol thus provides adapted and reproducible biochemical methodologies to study the ATXN3 aggregation involved in SCA3 pathogenesis, thereby contributing the evaluation of potential therapeutic outcomes for this still incurable disease.

## Supporting information

**S1 Fig. SDS-PAGE allows a quantitative study of SDS-resistant ATXN3 aggregates.** (A) HEK-239T were transfected with the indicated plasmid amounts to express GFP-ATXN3 WT. After 48h, cells were lysed and total cell extracts were analysed by SDS-PAGE with the indicated antibody. (B) HEK-239T were transfected with the GFP-ATXN3 polyQ-expressing plasmid and collected 7 days post-transfection. Cells were lysed and fractionated as described in the methods section, except for the last step. After the last wash and centrifugation, the insoluble pellet was resuspended in various buffers: the insoluble buffer containing 4% SDS, a DTT buffer (50mM Tris 7.4, 150 mM NaCl, 100 mM DTT), a Urea buffer (30 mM Tris pH 8.5, 7 M urea, 2 M thiourea) and a mix buffer (30 mM Tris pH 8.5, 7 M urea, 2 M thiourea, 100 mM DTT, 4% SDS). Subsequently, soluble and insoluble fractions were analysed by SDS-PAGE and membranes were immunoblotted for GFP. (C) HEK-239T transfected with GFP-ATXN3 polyQ, were lysed 7 days after transfection, fractionated and run on SDS-PAGE before being immunoblotted with the indicated antibodies. Electrophoresis was performed with 2 different voltages. (D) Purified His-tagged ATXN3-WT protein and extracts from cells expressing GFP-ATXN3 polyQ for 7 days were fractionated using the protocol described in the methods. Soluble and insoluble fractions were analysed by western blot with an antibody raised against the C-terminus of ATXN3 (E) Increasing amounts of insoluble fractions of cells expressing

GFP-ATXN3 polyQ for 7 days were analysed by SDS-PAGE. The GFP signal from the stacking gel of 3 replicates was quantified, and a linear correlation was applied between the quantity of protein loaded and the GFP signal. The GFP signal in the stacking gel was measured using ImageJ and quantifications were plotted against the amount of protein loaded.
(TIF)

**S2 Fig. Detecting ATXN3 aggregate formation by SDS-PAGE using various antibodies.** (A) HEK-239T expressing ATXN3 polyQ were lysed and fractionated at the indicated time points, followed by analysis by SDS-PAGE and immunoblot for GFP. (B) HEK-239T were transfected with GFP-ATXN3 polyQ expressing plasmid and cells were collected at the indicated times. After lysis and fractionation, samples were analysed by western blot using the indicated antibodies. (C) HEK-239T were transfected with siRNAs against luciferase (siLUC) or ATXN3 (siATXN3) for 48 hours, after which whole cell extracts were prepared. The extracts were analysed together with fractionated samples from HEK-239T transfected with GFP-ATXN3 polyQ expressing plasmid for 7 days by SDS-PAGE and western blot using the indicated antibodies. The asterisk marks the mobility of endogenous ATXN3.
(TIF)

**S3 Fig. FTA as a quantitative method to study of SDS-resistant ATXN3 aggregates.** (A) HEK-239T were transfected with the GFP-ATXN3 polyQ-expressing plasmid and collected 7 days post-transfection, followed by lysis and fractionation as described in the protocol section, except for the last step. After the last wash and centrifugation, the insoluble pellet was resuspended with various buffers, including the insoluble buffer containing 4% SDS, a DTT buffer (50mM Tris 7.4, 150 mM NaCl, 100 mM DTT), an urea buffer (30 mM Tris pH 8.5, 7 M urea, 2 M thiourea) and a mix buffer (30 mM Tris pH 8.5, 7 M urea, 2 M thiourea, 100 mM DTT, 4% SDS). Soluble and insoluble fractions were subsequently analysed by FTA and immunoblotting for GFP. (B) HEK-239T expressing GFP-ATXN3 polyQ for 7 days were lysed, fractionated and analysed by FTA and immunoblot using a GFP antibody. The GFP signal in the insoluble fractions was quantified with ImageJ and plotted against the amount of protein loaded.
(TIF)

**S4 Fig. Boiling soluble fractions induced GFP signal in the stacking gel of SDS-PAGE.** HEK-239T were transfected with the GFP-ATXN3 polyQ and WT expressing plasmids and collected at day 7 post-transfection followed by lysis, fractionation and analysis by SDS-PAGE as described in the protocol section, except that in this case samples were left untreated or boiled before loading onto the acrylamide gel, as indicated.
(TIF)

**S1 File.**
(PDF)

**S1 Raw images.**
(PDF)

## Acknowledgments

We thank Dr. Veronique Smits for critical reading of the manuscript.

## Author Contributions

**Conceptualization:** Grégoire Quinet, Raimundo Freire.

**Funding acquisition:** Grégoire Quinet, Raimundo Freire.

**Investigation:** Grégoire Quinet, María Cristina Paz-Cabrera.

**Methodology:** Grégoire Quinet, María Cristina Paz-Cabrera.

**Supervision:** Raimundo Freire.

**Writing – original draft:** Grégoire Quinet.

**Writing – review & editing:** Raimundo Freire.

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
