## [Decision Letter · Decision Letter 0]

5 Aug 2024

PONE-D-24-24197Systematic biochemical analysis to study wild-type and polyglutamine expanded ATXN3 species in vivo.PLOS ONE

Dear Dr. Freire,

Thank you for submitting your manuscript to PLOS ONE. After careful consideration, we feel that it has merit but does not fully meet PLOS ONE’s publication criteria as it currently stands. Therefore, we invite you to submit a revised version of the manuscript that addresses the points raised during the review process.

Please play close attention to the suggestion to utilize anti-atxn3 antibodies instead of only anti-gfp. A revised manuscript should address this point.

We look forward to receiving your revised manuscript.

Kind regards,

Elise A. Kikis

Academic Editor

PLOS ONE

Journal requirements: 1. When submitting your revision, we need you to address these additional requirements. Please ensure that your manuscript meets PLOS ONE's style requirements, including those for file naming. The PLOS ONE style templates can be found at https://journals.plos.org/plosone/s/file?id=wjVg/PLOSOne_formatting_sample_main_body.pdf and https://journals.plos.org/plosone/s/file?id=ba62/PLOSOne_formatting_sample_title_authors_affiliations.pdf. 2. PLOS ONE now requires that authors provide the original uncropped and unadjusted images underlying all blot or gel results reported in a submission’s figures or Supporting Information files. This policy and the journal’s other requirements for blot/gel reporting and figure preparation are described in detail at https://journals.plos.org/plosone/s/figures#loc-blot-and-gel-reporting-requirements and https://journals.plos.org/plosone/s/figures#loc-preparing-figures-from-image-files. When you submit your revised manuscript, please ensure that your figures adhere fully to these guidelines and provide the original underlying images for all blot or gel data reported in your submission. See the following link for instructions on providing the original image data: https://journals.plos.org/plosone/s/figures#loc-original-images-for-blots-and-gels.   In your cover letter, please note whether your blot/gel image data are in Supporting Information or posted at a public data repository, provide the repository URL if relevant, and provide specific details as to which raw blot/gel images, if any, are not available. Email us at plosone@plos.org if you have any questions. 3. We note that the grant information you provided in the ‘Funding Information’ and ‘Financial Disclosure’ sections do not match.  When you resubmit, please ensure that you provide the correct grant numbers for the awards you received for your study in the ‘Funding Information’ section. 4. Thank you for stating the following financial disclosure:  [This work was supported by grants PID2022-139691OB-I00 funded by MCIN/AEI/10.13039/501100011033 and ERDF A way of making Europe (European Union) to RF and grant OA23/071 from Fundación DISA to GQ.].  Please state what role the funders took in the study.  If the funders had no role, please state: ""The funders had no role in study design, data collection and analysis, decision to publish, or preparation of the manuscript."" If this statement is not correct you must amend it as needed. Please include this amended Role of Funder statement in your cover letter; we will change the online submission form on your behalf. 5. We note that you have included the phrase “data not shown” in your manuscript. Unfortunately, this does not meet our data sharing requirements. PLOS does not permit references to inaccessible data. We require that authors provide all relevant data within the paper, Supporting Information files, or in an acceptable, public repository. Please add a citation to support this phrase or upload the data that corresponds with these findings to a stable repository (such as Figshare or Dryad) and provide and URLs, DOIs, or accession numbers that may be used to access these data. Or, if the data are not a core part of the research being presented in your study, we ask that you remove the phrase that refers to these data. 6. We are unable to open your Figure file [Fig1.eps, Fig2.eps, Fig3.eps
, Fig4.eps, Fig5.eps and supplementary S1, S2]. Please kindly revise as necessary and re-upload.

Additional Editor Comments:

Please address the reviewers’ concerns in a revised manuscript, I encourage you to especially note the suggestion to utilize ATXN3-specific antibodies instead of only anti-gfp antibodies.

Reviewers' comments:

Reviewer's Responses to Questions

**Comments to the Author**

1. Does the manuscript report a protocol which is of utility to the research community and adds value to the published literature?

Reviewer #1: Yes

Reviewer #2: Yes

2. Has the protocol been described in sufficient detail?

To answer this question, please click the link to protocols.io in the Materials and Methods section of the manuscript (if a link has been provided) or consult the step-by-step protocol in the Supporting Information files.

The step-by-step protocol should contain sufficient detail for another researcher to be able to reproduce all experiments and analyses.

Reviewer #1: Yes

Reviewer #2: Partly

3. Does the protocol describe a validated method?

Reviewer #1: Yes

Reviewer #2: No

4. If the manuscript contains new data, have the authors made this data fully available?

Reviewer #1: N/A

Reviewer #2: No

**5. Is the article presented in an intelligible fashion and written in standard English?**

Reviewer #1: Yes

Reviewer #2: Yes

6. Review Comments to the Author

Reviewer #1: In the manuscript entitled “Systematic biochemical analysis to study wild-type and polyglutamine expanded ATXN3 species in vivo”, Quinet et al. outline a multi-step protocol to evaluate the aggregation stages of the aggregation prone protein ATXN3 in its wild-type and mutant form. To assess aggregation, the authors describe a step by step protocol employing commonly used strategies to evaluate aggregation states. Using their pipeline, researchers are able to identify and evaluate the proposed four-step mechanism for the aggregation of ATXN3 in vivo. This mechanism includes an early and late oligomerization phase followed by protofibril then fibril formation. This reviewer agrees that protocols for characterizing the various stages of aggregation prone proteins from an in vivo system are important for developing strategies to reverse or eliminate aggregation in disease. Recommendations for timing post-transfection are made along with buffer conditions to resolve various species of ATXN3. However, clarification of the timing and quantitation aspect of the pipeline along with additional representative datasets are suggested before publication.

In Figure 2, the authors use SDS-PAGE to resolve ATXN3 states (monomers, polymers/modified forms, and HMW fibrils). Based on the representative dataset, there is a notable difference in the amount of protein in the soluble fraction for the GFP-ATXN3 (WT) and GFP-ATXN3 (polyQ) at day 2 that is not discussed. In line 205, the authors discuss levels of ATXN3 aggregation in the intermediate/oligomer state in the soluble fractions at the longer time point but do not offer an explanation for the different relative amounts at day 2. Is this possibly due to the turnover of GFP-ATXN3 (WT) early (day 2) vs late (day 7)? Is this dependent on the proteosome? The authors state in lines 343-344 that optimization of protein expression time is required to study WT and polyQ-expanded ATXN3 aggregation in vivo. A discussion that includes clarification on this point would strengthen the timing aspect of the study.

In line 245, the authors propose that SDS-PAGE with the adapted settings is a method that can be used to quantify different ATXN3 variants. Furthermore, in line 288, the authors state that SDS-PAGE allows a global quantification of SDS-insoluble HMW of ATXN3 polyQ. And finally, in lines 346-347, the authors conclude that SDS-PAGE can be used as a quantitative method for “all kinds” of ATXN3 forms. However, they do not offer a quantitative analysis for the Figure 2 dataset to support these statements. Adding this analysis would strengthen the quantitative aspect of the study.

Reviewer #2: The manuscript by Freire and coworkers describes several biochemical approaches to investigate ATXN3 with and without polyglutamine expansion in cultured cells. Expanded ATXN3 causes spinocerebellar ataxia type 3 (also known as Machado-Joseph disease), an inherited neurodegenerative disease that is associated with misfolding and aggregation of the mutant protein, and a comprehensive summary of available methods to probe the various states of ATXN3 misfolding would be a valuable addition to the field. The manuscript focuses on three assays, filter trap, SDS-polyacrylamide and SDS-agarose gel electrophoresis, to visualize different species of GFP-tagged ATXN3 protein. These methods have been used extensively by researchers to explore misfolding of ATXN3 and other polyglutamine proteins such as huntingtin over the last 20+ years.

Major comments:

1- My main concern is related to the fact that the authors define monomers, oligomers/polymers/modified forms, and high molecular weight fibrils or aggregates based on electrophoretic shift alone. The results would have been more convincing if the same reference standards (such as purified monomeric ATXN3 and in vitro assembled fibrils) were included to firmly establish the identity of the species and compare the three methods.

2- Another issue is that all three assays employ antibodies that recognize the GFP tag to detect ATXN3 protein. Assuming that the authors transfected commonly used plasmids encoding ATXN3 with an amino-terminal GFP tag (Addgene plasmid #22122 and #22123), such an indirect detection via the GFP tag would miss any fragments containing the carboxy-terminal polyglutamine tract. This is particularly important considering proteolytic processing of ATXN3, which the authors state “plays a crucial role in its toxicity” [p. 7 line 213]. Therefore, all immunoblots should be probed with multiple antibodies targeting other regions in the ATXN3 protein, including antibody clone 1C2, which specifically recognizes the expanded polyglutamine tract (see for example Song et al. 2022 Life Med 1:27-44; and Merry et al. 1998 Hum Mol Genet 7:693-701).

3- It is unclear how the timepoints (1 and 7 days after transfection) were chosen for analysis. HEK293T cells have a doubling time of roughly ~24 hours; waiting one week after the transfection to collect samples seems a bit long. When does misfolding/aggregation start under these conditions? Is it a gradual process?

4- The text states that “multi-step aggregation of ATXN3 is time-dependent and can be influenced by several parameters, including the biological model used, the type of ATXN3 isoforms and the amount of expressed proteins” [p. 4 line 93], yet is remains unclear how parameters were chosen for this particular protocol. This needs to be clarified – ideally with data.

5- The title does not accurately reflect the contents of the manuscript. I recommend removing the terms “Systematic” (see comments above) and “in vivo” (the work was done using protein extracts from HEK293T cells).

Minor comments:

6- The protocol does not include a description of the plasmid DNA used for transfection. This is important, as the extent of protein expression will depend on the promoter used. This information needs to be added.

7- Step 5 of the protocol states “renew medium if it turns yellow”. The methods section should clarify how often the cell culture medium needs to be changed after the transfection.

7. PLOS authors have the option to publish the peer review history of their article (what does this mean?). If published, this will include your full peer review and any attached files.

Reviewer #1: No

Reviewer #2: No

---

## [Author Response · Author response to Decision Letter 0]

4 Oct 2024

--

We included all the raw data from the blots presented throughout the manuscript in Supporting Information (S1 raw images file).

The work was supported by 2 grants: 

1- PID2022-139691OB-I00 from the Agencia Estatal de Investigación (which includes funds from the European Union)

2- OA23/071 from Fundación DISA.

According to the instructions, funding from the Agencia Estatal de Investigación acknowledged in the following way: Funded by MCIN/AEI/10.13039/501100011033 and ERDF A way of making Europe (European Union)”. In the initial submission it was not possible to add all this information in the “Funding information” section, but was added in the “Financial Disclosure”. Since you require the two sections to be the same, we corrected the “Financial Disclosure” to a shorter version. However, is very important that the funding information in the article itself contains the correct, longer version and we therefore left that information in the “Funding” section in the article. 

[This work was supported by grants PID2022-139691OB-I00 funded by MCIN/AEI/10.13039/501100011033 and ERDF A way of making Europe (European Union) to RF and grant OA23/071 from Fundación DISA to GQ.]. 

The role of the funders in the study (no role) is now added to the funding information of the manuscript.

Data supporting the previous “data not shown” is now provided in S3 FigC. The text in the manuscript has been adapted accordingly (lines 360 to 364). 

6. We are unable to open your Figure file [Fig1.eps, Fig2.eps, Fig3.eps , Fig4.eps, Fig5.eps and supplementary S1, S2]. Please kindly revise as necessary and re-upload.

We are now uploading the figure files in .tiff format. Hopefully now all files can be opened. 

Thank you for submitting your manuscript to PLOS ONE. After careful consideration, we feel that it has merit but does not fully meet PLOS ONE’s publication criteria as it currently stands. Therefore, we invite you to submit a revised version of the manuscript that addresses the points raised during the review process.

Please play close attention to the suggestion to utilize anti-atxn3 antibodies instead of only anti-gfp. A revised manuscript should address this point.

This revised version of the manuscript contains new experiments suggested by the reviewers, including the use of (3) new, different antibodies to detect ATXN3, as highlighted by the editor. The new manuscript therefore includes more figures/figure panels in addition to changes in the text, following the suggestions of reviewers. We hope this revised version of the manuscript now merits is publication in PLOS ONE.

Reviewer #1: In the manuscript entitled “Systematic biochemical analysis to study wild-type and polyglutamine expanded ATXN3 species in vivo”, Quinet et al. outline a multi-step protocol to evaluate the aggregation stages of the aggregation prone protein ATXN3 in its wild-type and mutant form. To assess aggregation, the authors describe a step by step protocol employing commonly used strategies to evaluate aggregation states. Using their pipeline, researchers are able to identify and evaluate the proposed four-step mechanism for the aggregation of ATXN3 in vivo. This mechanism includes an early and late oligomerization phase followed by protofibril then fibril formation. This reviewer agrees that protocols for characterizing the various stages of aggregation prone proteins from an in vivo system are important for developing strategies to reverse or eliminate aggregation in disease. Recommendations for timing post-transfection are made along with buffer conditions to resolve various species of ATXN3. However, clarification of the timing and quantitation aspect of the pipeline along with additional representative datasets are suggested before publication.

A 7-day time course to study the progression of aggregation in our experimental setup was performed. This experiment made us choose two time points for further experiments in the manuscript: day 2 and day 7 post-transfection, as they represent early and late times of the ATXN3 aggregation process, respectively. We modified the text explaining this (lines 185-189). 

In addition, as suggested by reviewer, new quantifications of 3 independent experiments were added in Figure 2B and Figure 3B.

In Figure 2, the authors use SDS-PAGE to resolve ATXN3 states (monomers, polymers/modified forms, and HMW fibrils). Based on the representative dataset, there is a notable difference in the amount of protein in the soluble fraction for the GFP-ATXN3 (WT) and GFP-ATXN3 (polyQ) at day 2 that is not discussed. In line 205, the authors discuss levels of ATXN3 aggregation in the intermediate/oligomer state in the soluble fractions at the longer time point but do not offer an explanation for the different relative amounts at day 2. Is this possibly due to the turnover of GFP-ATXN3 (WT) early (day 2) vs late (day 7)? Is this dependent on the proteosome? The authors state in lines 343-344 that optimization of protein expression time is required to study WT and polyQ-expanded ATXN3 aggregation in vivo. A discussion that includes clarification on this point would strengthen the timing aspect of the study.

The reviewer is right, thank you for pointing this out. We believe this effect is due to differences in transfection efficiency but indeed, also might be due to different protein stability of the two variants. We now discuss these possibilities in the manuscript (lines 214 to 218).

In line 245, the authors propose that SDS-PAGE with the adapted settings is a method that can be used to quantify different ATXN3 variants. Furthermore, in line 288, the authors state that SDS-PAGE allows a global quantification of SDS-insoluble HMW of ATXN3 polyQ. And finally, in lines 346-347, the authors conclude that SDS-PAGE can be used as a quantitative method for “all kinds” of ATXN3 forms. However, they do not offer a quantitative analysis for the Figure 2 dataset to support these statements. Adding this analysis would strengthen the quantitative aspect of the study.

Quantification and statistical analyses of three independent experiments were added in Figure 2B for SDS-PAGE and in Figure 3 for FTA. The text was modified accordingly (lines 264 to 267).

Reviewer #2: The manuscript by Freire and coworkers describes several biochemical approaches to investigate ATXN3 with and without polyglutamine expansion in cultured cells. Expanded ATXN3 causes spinocerebellar ataxia type 3 (also known as Machado-Joseph disease), an inherited neurodegenerative disease that is associated with misfolding and aggregation of the mutant protein, and a comprehensive summary of available methods to probe the various states of ATXN3 misfolding would be a valuable addition to the field. The manuscript focuses on three assays, filter trap, SDS-polyacrylamide and SDS-agarose gel electrophoresis, to visualize different species of GFP-tagged ATXN3 protein. These methods have been used extensively by researchers to explore misfolding of ATXN3 and other polyglutamine proteins such as huntingtin over the last 20+ years.

Major comments:

1- My main concern is related to the fact that the authors define monomers, oligomers/polymers/modified forms, and high molecular weight fibrils or aggregates based on electrophoretic shift alone. The results would have been more convincing if the same reference standards (such as purified monomeric ATXN3 and in vitro assembled fibrils) were included to firmly establish the identity of the species and compare the three methods. 

As suggested by the reviewer we expressed and purified a His-tagged ATXN3 WT version of the protein and performed the biochemical fractionation before analysis by SDS-PAGE. The purified ATXN3 WT was only detected in the soluble fraction and ran as a monomer, suggesting that the other ATXN3 forms observed after fractionation and SDS-PAGE analysis from cell extracts are products of different stages of aggregation. Although we failed to clone and express ATXN3 polyQ we believe that the data obtained with purified ATXN3 WT demonstrates that our protocol does not result in extra artifacts. These data was added as S1 FigD with an explanation in the text (and 162 to 167).

2- Another issue is that all three assays employ antibodies that recognize the GFP tag to detect ATXN3 protein. Assuming that the authors transfected commonly used plasmids encoding ATXN3 with an amino-terminal GFP tag (Addgene plasmid #22122 and #22123), such an indirect detection via the GFP tag would miss any fragments containing the carboxy-terminal polyglutamine tract. This is particularly important considering proteolytic processing of ATXN3, which the authors state “plays a crucial role in its toxicity” [p. 7 line 213]. Therefore, all immunoblots should be probed with multiple antibodies targeting other regions in the ATXN3 protein, including antibody clone 1C2, which specifically recognizes the expanded polyglutamine tract (see for example Song et al. 2022 Life Med 1:27-44; and Merry et al. 1998 Hum Mol Genet 7:693-701).

The same soluble and insoluble fractions of ATXN3-PolyQ at 2 and 7 days post transfection were analysed with 3 antibodies: one recognising the full length ATXN3, another recognising the C-terminal region of ATXN3 (amino acids 140-361) and the 5TF1-1C2 monoclonal antibody that recognizes polyQ, suggested by the reviewer. We obtained similar results with all antibodies, validating our results with the anti-GFP antibody (lines 229 to 245). The main difference is that with the new antibodies, extra low molecular weight forms were detected. As some of these bands move exactly as endogenous ATXN3, we are not sure that the antibodies recognize products of degraded GFP-ATXN3 polyQ or they recognize endogenous full length and processed ATXN3. Because of this we decided to use the anti-GFP antibody for the rest of the experiments in the manuscript. This matter is discussed in the revised manuscript (lines 229 to 245).

3- It is unclear how the timepoints (1 and 7 days after transfection) were chosen for analysis. HEK293T cells have a doubling time of roughly ~24 hours; waiting one week after the transfection to collect samples seems a bit long. When does misfolding/aggregation start under these conditions? Is it a gradual process?

As described in various studies (references 8-11) and explained in the introduction (lines 58 to 66, and line 79), aggregation is a time-dependent process. As previous experiments using our experimental setup indicated that the aggregation of ATXN3 polyQ occurs at 5-7 days after transfection, we performed a 7-day time course post transfection.

As shown in S2 FigA, a small fraction of insoluble fibrils can be observed at 3 days after transfection, with amounts increasing at 6-7 days post-transfection. We therefore harvested the cells for all experiments with the different techniques in this manuscript at day 2 (no SDS-resistant HMW ATXN3 polyQ aggregation detected), and at day 7 (maximal SDS-resistant HMW ATXN3 polyQ detected), as representative time points for early and late aggregation stages, respectively. 

Similar to our answer to reviewer 1 (comment 1), this choice is better explained in the revised manuscript (lines 185-189). 

4- The text states that “multi-step aggregation of ATXN3 is time-dependent and can be influenced by several parameters, including the biological model used, the type of ATXN3 isoforms and the amount of expressed proteins” [p. 4 line 93], yet is remains unclear how parameters were chosen for this particular protocol. This needs to be clarified – ideally with data.

The choice for a simple cellular model is explained in lines 113-114, supported by citations. After preliminary tests (included in the new S1 FigA) we decided to choose to transfect 3 ug of ATXN3 WT plasmid, as in this case the protein was easily detected, but not heavily overexpressed. The same experiments were performed with the ATXN3 polyQ expressing plasmid. Explanation is added to the text (line 117 to 121).

5- The title does not accurately reflect the contents of the manuscript. I recommend removing the terms “Systematic” (see comments above) and “in vivo” (the work was done using protein extracts from HEK293T cells).

As suggested, the title of the manuscript has been adapted.

Minor comments:

6- The protocol does not include a description of the plasmid DNA used for transfection. This is important, as the extent of protein expression will depend on the promoter used. This information needs to be added.

Information regarding the used plasmids is now included.

7- Step 5 of the protocol states “renew medium if it turns yellow”. The methods section should clarify how often the cell culture medium needs to be changed after the transfection.

This sentence was changed (step 5 of the protocol).

---

## [Decision Letter · Decision Letter 1]

15 Nov 2024

PONE-D-24-24197R1Biochemical analysis to study wild-type and polyglutamine-

expanded ATXN3 speciesPLOS ONE

Dear Dr. Freire,

Thank you for submitting your manuscript to PLOS ONE. After careful consideration, we feel that it has merit but does not fully meet PLOS ONE’s publication criteria as it currently stands. Therefore, we invite you to submit a revised version of the manuscript that addresses the points raised during the review process.

We look forward to receiving your revised manuscript.

Kind regards,

Elise A. Kikis

Academic Editor

PLOS ONE

Journal Requirements:

Additional Editor Comments (if provided):

The manuscript is much improved. Please address the one comment by reviewer #2.

Reviewers' comments:

Reviewer's Responses to Questions

**Comments to the Author**

1. Does the manuscript report a protocol which is of utility to the research community and adds value to the published literature?

Reviewer #1: Yes

Reviewer #2: Yes

2. Has the protocol been described in sufficient detail?

To answer this question, please click the link to protocols.io in the Materials and Methods section of the manuscript (if a link has been provided) or consult the step-by-step protocol in the Supporting Information files.

The step-by-step protocol should contain sufficient detail for another researcher to be able to reproduce all experiments and analyses.

Reviewer #1: Yes

Reviewer #2: Yes

3. Does the protocol describe a validated method?

Reviewer #1: Yes

Reviewer #2: Yes

4. If the manuscript contains new data, have the authors made this data fully available?

Reviewer #1: Yes

Reviewer #2: N/A

**5. Is the article presented in an intelligible fashion and written in standard English?**

Reviewer #1: Yes

Reviewer #2: Yes

6. Review Comments to the Author

Reviewer #1: I have answered all the questions above favorably. Thank you for carefully addressing reviewer comments.

Reviewer #2: Thank you for thoroughly addressing my previous comments. However, there is one point related to the new data that requires clarification. On page 9, line 236 the revised text states that “…two ATXN3 antibodies and the polyglutamine antibody recognized some proteins smaller than the GFP ATXN3 polyQ monomer, that were not recognized by the anti-GFP antibody (Fig 2A, S2 FigB). These bands could either correspond to overexpressed GFP-ATXN3 polyQ that was processed to smaller fragments without GFP or to (modified forms of) endogenous ATXN3.”

It is unclear which smaller bands the authors are referring to in Figure S2B; I assume it is the double-band in the soluble fraction, but this should be specified. Additionally, it seems unlikely that the smaller band corresponds to endogenous ATXN3, given the cross-reactivity with the 1C2 antibody, which recognizes expanded polyglutamine. Aside from this, my concerns have been addressed.

7. PLOS authors have the option to publish the peer review history of their article (what does this mean?). If published, this will include your full peer review and any attached files.

Reviewer #1: No

Reviewer #2: No

---

## [Author Response · Author response to Decision Letter 1]

27 Nov 2024

Journal Requirements:

Additional Editor Comments (if provided):

The manuscript is much improved. Please address the one comment by reviewer #2.

Reviewers' comments:

Reviewer's Responses to Questions

Reviewers' comments:

Reviewer's Responses to Questions

Comments to the Author

Reviewers' comments:

Reviewer's Responses to Questions

Comments to the Author

1. Does the manuscript report a protocol which is of utility to the research community and adds value to the published literature?

Reviewer #1: Yes

Reviewer #2: Yes

2. Has the protocol been described in sufficient detail?

To answer this question, please click the link to protocols.io in the Materials and Methods section of the manuscript (if a link has been provided) or consult the step-by-step protocol in the Supporting Information files.

The step-by-step protocol should contain sufficient detail for another researcher to be able to reproduce all experiments and analyses.

Reviewer #1: Yes

Reviewer #2: Yes

3. Does the protocol describe a validated method?

Reviewer #1: Yes

Reviewer #2: Yes

4. If the manuscript contains new data, have the authors made this data fully available?

Reviewer #1: Yes

Reviewer #2: N/A

5. Is the article presented in an intelligible fashion and written in standard English?

Reviewer #1: Yes

Reviewer #2: Yes

6. Review Comments to the Author

Reviewer #1: I have answered all the questions above favorably. Thank you for carefully addressing reviewer comments.

We thank the reviewer for his/her comments to help to improve our manuscript.

Reviewer #2: Thank you for thoroughly addressing my previous comments. However, there is one point related to the new data that requires clarification. On page 9, line 236 the revised text states that “…two ATXN3 antibodies and the polyglutamine antibody recognized some proteins smaller than the GFP ATXN3 polyQ monomer, that were not recognized by the anti-GFP antibody (Fig 2A, S2 FigB). These bands could either correspond to overexpressed GFP-ATXN3 polyQ that was processed to smaller fragments without GFP or to (modified forms of) endogenous ATXN3.”

It is unclear which smaller bands the authors are referring to in Figure S2B; I assume it is the double-band in the soluble fraction, but this should be specified. Additionally, it seems unlikely that the smaller band corresponds to endogenous ATXN3, given the cross-reactivity with the 1C2 antibody, which recognizes expanded polyglutamine. Aside from this, my concerns have been addressed.

Thanks to the reviewer for his/her comments. To address this last comment, we added new panels in supplemental figure 2 (S2 Fig C) in which we analysed total extracts of control and ATXN3-depleted cells together with the fractionation of GFP-ATXN3 polyQ expressing cells by western blot, using the two ATXN3 antibodies used in S2 Fig B. The double band that corresponds to endogenous ATXN3, was marked with an asterisk in the figure S2 for clarification. This double band migrated at the same mobility as bands present after fractionation of cells expressing GFP-ATXN3 polyQ. Therefore, we find it likely that these bands correspond to the endogenous ATXN3 protein. Accordingly, new text was added (lines 239-242), the S2 figure legend was adapted (lines 423-428) in the main manuscript and information about the siRNA oligonucleotides was included in the S1 file.

7. PLOS authors have the option to publish the peer review history of their article (what does this mean?). If published, this will include your full peer review and any attached files.

Do you want your identity to be public for this peer review? For information about this choice, including consent withdrawal, please see our Privacy Policy.

Reviewer #1: No

Reviewer #2: No

---

## [Editor Report · Decision Letter 2]

3 Dec 2024

Biochemical analysis to study wild-type and polyglutamine-expanded ATXN3 species

PONE-D-24-24197R2

Dear Dr. Freire,

We’re pleased to inform you that your manuscript has been judged scientifically suitable for publication and will be formally accepted for publication once it meets all outstanding technical requirements.

Kind regards,

Elise A. Kikis

Academic Editor

PLOS ONE
---

## [Editor Report · Acceptance letter]

11 Dec 2024

PONE-D-24-24197R2 

PLOS ONE

Dear Dr. Freire, 

I'm pleased to inform you that your manuscript has been deemed suitable for publication in PLOS ONE. Congratulations! Your manuscript is now being handed over to our production team.

Kind regards, 

on behalf of

Dr. Elise A. Kikis 

Academic Editor

PLOS ONE